# Hydrogen Peroxide Interference in Chemical Oxygen Demand Assessments of Plasma Treated Waters

**Joseph Groele [1],\* and John Foster [2]**

[1] Department of Mechanical Engineering, University of Michigan, Ann Arbor, MI 48109, USA
[2] Department of Nuclear Engineering and Radiological Sciences, University of Michigan, Ann Arbor, MI 48109, USA
\* Correspondence: jrgroele@umich.edu; Tel.: +1-716-307-5651

**Abstract:** Plasma-driven advanced oxidation represents a potential technology to safely re-use waters polluted with recalcitrant contaminants by mineralizing organics via reactions with hydroxyl radicals, thus relieving freshwater stress. The process results in some residual hydrogen peroxide, which can interfere with the standard method for assessing contaminant removal. In this work, methylene blue is used as a model contaminant to present a case in which this interference can impact the measured chemical oxygen demand of samples. Next, the magnitude of this interference is investigated by dosing de-ionized water with hydrogen peroxide via dielectric barrier discharge plasma jet and by solution. The chemical oxygen demand increases with increasing concentration of residual hydrogen peroxide. The interference factor should be considered when assessing the effectiveness of plasma to treat various wastewaters.

**Keywords:** non-equilibrium plasma applications; water treatment; advanced oxidation

## 1. Introduction

Demand for freshwater is rapidly rising due to expanding agriculture and industrialization; meanwhile, population growth, changing climate, accidental spills, and deteriorating infrastructure further exacerbates the problem of freshwater scarcity [1]. One possible approach to managing this impeding crisis is to re-use wastewater by removing contaminants; however, wastewaters contain contaminants of emerging concern (CECs) which are not readily removed by traditional water treatment processes. These CECs include pharmaceuticals, industrial chemicals such as per- and polyfluoroalkyl substances (PFAS), pesticides for agriculture, and microcystins [2–6]. Fortunately, these CECs can be removed from wastewaters using advanced oxidation processes [7].

Advanced oxidation processes (AOPs) are a category of chemical treatment methods for removing persistent organic pollutants from waters and wastewaters via reactions with highly reactive oxidizing agents, namely hydroxyl radicals. Traditional AOPs typically leverage combinations of ozone, hydrogen peroxide, and ultraviolet light to facilitate in-situ generation of hydroxyl radicals for non-selective decomposition of organic contaminants [8,9]. Through a series of chain reactions known as mineralization, hydroxyl radicals react with organics to ultimately form carbon dioxide, water, and mineral ions, with aldehydes and carboxylic acids often serving as intermediate breakdown products [10].

Plasma-liquid interaction can generate hydrogen peroxide, ozone, and other reactive species along with ultraviolet radiation and short-lived highly reactive hydroxyl radicals, thus representing a novel AOP for the removal of recalcitrant organic compounds from waters and wastewaters [11–13]. The formation of these reactive species is initiated primarily through energetic electrons from the plasma region colliding with atoms or molecules, either in the gas phase or at the liquid surface.

Subsequent production of reactive species can also be facilitated by recombination of radicals and de-excitation of metastables [14].

Plasma discharges in contact with water can generate multitudes of reactive oxygen species, including superoxide, hydroperoxyl, and atomic oxygen. If the discharge is in air, then reactive nitrogen species will also form, including nitric oxide, nitrite, nitrate, and peroxynitrate [15]. While all of these reactive oxygen and nitrogen species may contribute to degradation of contaminants in wastewaters, the most important species for advanced oxidation are hydroxyl radicals, hydrogen peroxide, and ozone. In particular, hydroxyl radicals react non-selectively with organics, including contaminant intermediate products such as short chain carboxylic acids, to allow for complete mineralization of most organic pollutants [16].

The production of hydroxyl radicals in plasmas in or in contact with water can occur through a number of different pathways, with the relative importance of these pathways depending on the plasma parameters and the gas composition [17]. Some of these hydroxyl radicals produced in the discharge diffuse to the liquid surface and are transported across the gas-water interface to either react with pollutants in the water or scavenge themselves to form longer lived hydrogen peroxide [12,18].

In the bulk liquid, hydrogen peroxide can react with any dissolved ozone from the discharge to produce more hydroxyl radicals (i.e., peroxone process [19]) for in-situ oxidation of organics, even after the plasma has been turned off. However, not all of the hydrogen peroxide is consumed through reactions with ozone, and a portion of the post-discharge hydroxyl radicals produced by the peroxone process will dimerize back into hydrogen peroxide. In this way, plasma-driven advanced oxidation can leave residual hydrogen peroxide in treated waters that may persist for days, as can also happen with traditional AOPs [20].

The goal of plasma-driven advanced oxidation, and AOPs in general, is to chemically remove contaminants from waters. The five most prominent methods for assessing contaminant removal are liquid chromatography tandem mass spectrometry (LC-MS), spectrophotometry, total organic carbon (TOC), biochemical oxygen demand (BOD), and chemical oxygen demand (COD) [21–24]. Spectrophotometry and LC-MS allow for the determination of specific species concentrations, whereas TOC, BOD, and COD are non-specific water quality parameters that provide a measurement of the overall pollution potential of a water sample. When initially investigating the performance of AOPs for removal of contaminants from wastewaters, LC-MS should be used to validate results from other methods; however, for general monitoring of effluent quality in water and wastewater treatment processes, TOC, BOD, and COD are more commonly used due to the simplicity of the test procedure and easily interpreted results.

As the name suggests, TOC is a measure of the total organic carbon contained in a water sample. This measurement involves two stages: total carbon (TC) and inorganic carbon (IC) measurements, with the difference providing the TOC of the sample. First, combustion catalytic oxidation of the sample converts the organic carbon to carbon dioxide, which is then cooled and humidified before being detected by an infrared gas analyzer to measure the TC content of the sample. The second step involves acidifying the sample to pH < 4, making bicarbonate and carbonate unstable [21,25]. Thus, the IC in the sample is converted to carbon dioxide, either free $CO_{2(aq)}$ or in the form of carbonic acid, that can be purged with a $CO_2$-free gas to isolate the carbon dioxide, which is subsequently cooled, humidified, and detected by the gas analyzer as the IC measurement.

While TOC measurements focus on the carbon content of a sample directly, BOD and COD are measurements of the amount of oxygen consumed in the complete oxidation of organics to carbon dioxide, water, and mineral salts. As such, BOD and COD are particularly well suited for monitoring water treatment processes involving oxidation of organics and for informing the design of the water treatment process (e.g., how much oxidant must be used to remove the contaminants present in the waste stream).

More specifically, BOD is a measure of the oxygen consumed by bacteria in the oxidation of organics and is representative of contaminant decomposition in the natural environment, making this

method most suitable for predicting the effects of the organic contaminants on the dissolved oxygen levels of receiving waters. In contrast, COD is a measure of the oxygen consumed in the chemical oxidation of organics [23]. The BOD test typically takes five days to get results, meanwhile the COD test can provide results in less than three hours.

Due to the short analysis time and ability to correlate with BOD, the COD test has become the industry standard for rapid and frequent monitoring of water treatment process efficiency and effluent quality. The most common method is the closed reflux, colorimetric method with potassium dichromate oxidizing agent [22]. Pre-formulated commercial reagent mixtures are available for rapid COD analysis and provide consistent results between different laboratories. The problem with COD assessments of plasma treated wastewaters is that residual peroxide interferes with the measurement. Indeed, this interference is an issue for any AOP involving residual peroxide [20]. This paper first discusses a case in which this interference can lead to misinterpretation of plasma treatment results, and then investigates the magnitude of the interference in COD assessments of waters containing residual hydrogen peroxide.

## 2. Materials and Methods

### 2.1. Underwater Dielectric Barrier Discharge Plasma Jet

The discharge configuration used to bring non-equilibrium plasma in contact with water in this investigation is an underwater dielectric barrier discharge (DBD) plasma jet. The set-up is shown in Figure 1 and is based on the design by Foster et al. [26]. The plasma applicator consists of an 18 gauge tin-plated copper high-voltage wire centered coaxially within a quartz tube with 4 mm ID, 6.35 mm OD, and 150 mm length, and a tin-plated copper wire coil wrapped around the outside of the quartz tube to serve as the grounded electrode. In this case, the quartz acts as the dielectric barrier preventing a thermal arc from forming.

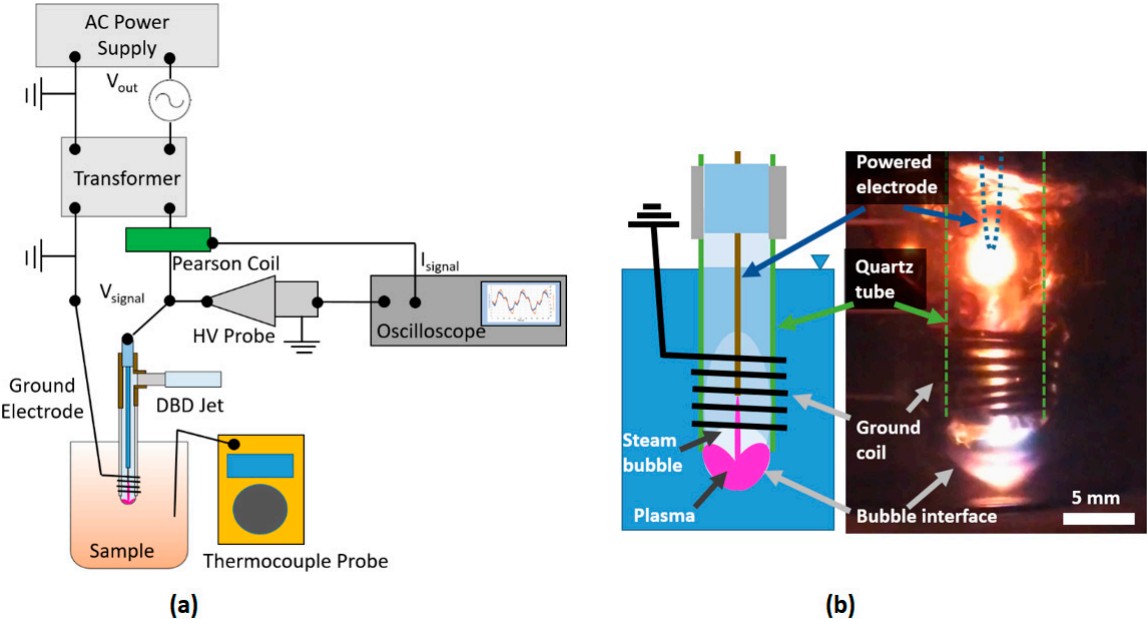

**Figure 1.** (**a**) Schematic of the DBD plasma jet experimental set-up used in this work. (**b**) Schematic and image of the DBD plasma jet apparatus operating in steam-mode discharge in de-ionized water.

The central high-voltage electrode is excited with a 5 kHz sinusoidal voltage at 4 $kV_{p-p}$, provided by an Elgar model 501SL power supply (AMETEK, Inc., Berwyn, PA, USA) with a 50:1 step-up transformer (SP-225 Plasma Technics, Inc., Racine, WI, USA). Voltage was measured using a high-voltage probe (P6015A, Tektronix, Beaverton, OR, USA), and the discharge current was measured with a Pearson coil

current monitor (6585, Pearson Electronics, Inc., Palo Alto, CA, USA). Current and voltage waveforms were recorded using a 2 GHz oscilloscope (Wavepro 7200a, Teledyne LeCroy, Chestnut Ridge, NY, USA). Typical current-voltage data is shown in Figure 2. Data from previous Lissajous experiments by Garcia et al. with the DBD plasma jet operating in steam-mode at these power supply conditions indicate that about 56 W of power are deposited in the discharge, with peak steam temperatures around 2800 K, as determined from comparing theoretical simulation of OH(A-X) spectra with experimental optical emission spectra [27].

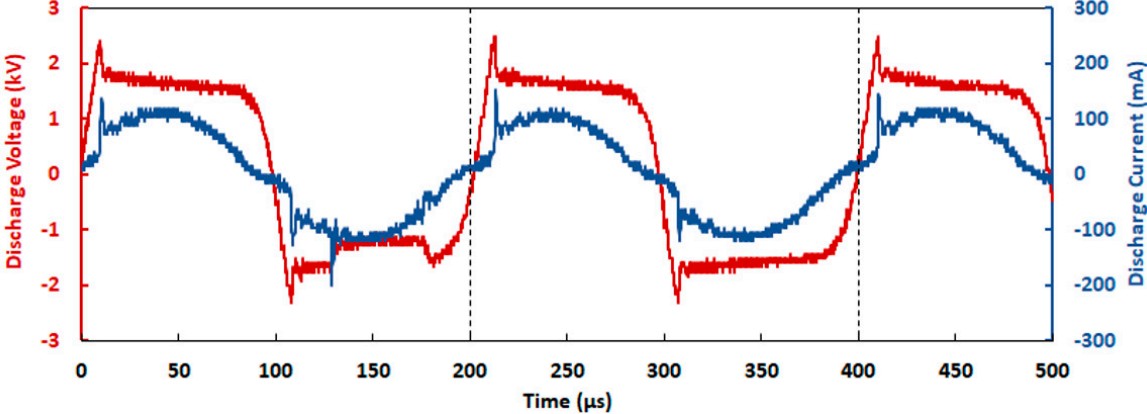

**Figure 2.** Discharge voltage and current signals measured during operation of the underwater DBD plasma jet operating in steam-mode at 5 kHz.

This DBD plasma jet can be operated with or without gas flow. In this work, the plasma jet operates without any input gas. This so-called "steam mode" discharge relies on local evaporation near the tip of the high-voltage electrode to form a vapor bubble that acts as the low-density region for plasma formation. Although a portion of the deposited power goes into heating and evaporating water, operating in steam-mode requires no consumables and minimizes the production of $NO_x$, as evidenced from optical emission spectroscopy [28,29], which can interfere with the iodometric titration method for quantifying hydrogen peroxide concentration, discussed in Section 2.3.

### 2.2. Sample Preparation

Samples of de-ionized water (Milli-Q EMD Millipore, Burlington, MA, USA, electrical conductivity ≈ 3 μS/cm) are dosed with hydrogen peroxide either by solution (30%, Fisher Chemical, Hampton, NH, USA) using a variable volume sampler pipette (Thermo Fisher Scientific, Waltham, MA, USA, resolution: 1 μL) or by underwater DBD plasma jet. When dosing with the DBD plasma jet, the opening of the quartz tube is placed approximately in the center of a 50 mL sample of de-ionized water and power is provided to the central high-voltage electrode. Approximately one second is required for the vapor bubble to form at the electrode tip in which the plasma discharge subsequently develops.

Plasma is applied to the sample via the DBD plasma jet for 30 s at a time, and the increase in sample temperature throughout the treatment duration is measured using a Fluke 87V true RMS multimeter with a type-K thermocouple (Fluke Corporation, Everett, WA, USA, resolution: 0.1 °C, accuracy: 0.05%). After 30 s of treatment, the sample is placed in an ice bath to cool down to room temperature before further treatment. The objective was to keep the bulk liquid sample below 60 °C. After treatment, the samples are stored in a dark cabinet for approximately 20 h before the hydrogen peroxide concentration and COD are measured to allow post-discharge reactions to take place while mitigating hydrogen peroxide photo-dissociation.

Samples of methylene blue (MB) are prepared by dissolving solid MB powder (M291-25 Fisher Chemical) in de-ionized water (Milli-Q EMD Millipore). A low concentration sample of 5 mg/L and a high concentration sample of 1000 mg/L are prepared and treated with the underwater DBD plasma jet

to demonstrate the hydrogen peroxide interference problem, discussed in Section 3. Plasma treatment of the MB samples follows the same procedure as described for dosing de-ionized water samples with hydrogen peroxide by DBD plasma jet, including monitoring the temperature to keep the sample below 60 °C. Again, treated MB samples were stored in a dark cabinet for 20 h before assessing the COD.

### 2.3. Hydrogen Peroxide Measurement

The hydrogen peroxide concentration in the sample is quantified using the iodometric titration method (Hach test kit model HYP-1, Hach Company, Loveland, CO, USA, resolution: 1 mg/L $H_2O_2$) immediately prior to COD assessment. The titration method involves the oxidation of iodide to iodine (Equation (1)) in the presence of a molybdate catalyst and a starch indicator. The starch indicator forms a dark blue complex with iodine. Subsequent titration with thiosulfate under acidic conditions (approximate pH of 4) converts the iodine back to iodide (Equation (2)), and the color change from blue to transparent indicates the titration endpoint.

$$H_2O_2 + 2KI + H_2SO_4 \rightarrow I_2 + K_2SO_4 + 2H_2O \tag{1}$$

$$I_2 + 2Na_2S_2O_3 \rightarrow Na_2S_4O_6 + 2NaI \tag{2}$$

$$2I^- + 2NO_2^- + 4H^+ \rightarrow I_2 + 2NO + 2H_2O \tag{3}$$

This iodometric titration method is subject to interference from nitrite ions (Equation (3)); thus, nitrogen species from air plasma discharges will interfere with this measurement. To mitigate the generation of reactive nitrogen species that could result in aqueous nitrite, the DBD plasma jet is operated in steam mode, as discussed in Section 2.1. For future studies, colorimetric assay by titanyl sulfate reagent with azide for elimination of nitrite interference is a preferred method of hydrogen peroxide quantification due to superior selectivity, making it suitable for hydrogen peroxide concentration measurements in waters treated with air plasmas [15,30].

### 2.4. Chemical Oxygen Demand Measurement

The assessment of COD in this work was performed using the USEPA 4.10.4 approved method, which is the closed reflux, colorimetric method with potassium dichromate oxidizer. Pre-formulated reagent was purchased from Hanna Instruments (HI9375A-25 COD reagent low range: 0 to 150 mg/L as $O_2$, accuracy: ±5 mg/L or ±4% of reading at 25 °C, resolution: 1 mg/L) along with a Hanna Instruments photometer (HI83399, Hanna Instruments, Woonsocket, RI, USA) for measuring sample absorbance using an LED light source with narrow band interference filter at 420 nm.

## 3. Results and Discussion

To illustrate a picture of the problem, a 100 mL sample of 5 mg/L MB is prepared and the COD is assessed before and after treatment. The results are shown in Table 1. In general, the COD is expected to decrease with plasma treatment time as the MB molecules are mineralized. Indeed, the color of the dye disappears after 15 min of treatment indicating the destruction of MB, as seen in Figure 3a; yet, the measured COD actually increases with treatment time. No additional organics are being added to the solution by the plasma; thus, one of the plasma-derived species being transported to the liquid during treatment must be contributing to the measured COD of the sample.

**Table 1.** Measured COD values for untreated and treated samples of MB. MB-A samples (left) contain initial MB concentration of 5 mg/L while MB-B samples (right) start with 1000 mg/L MB.

| MB-A Sample | COD (mg/L) | MB-B Sample | COD (mg/L) |
|---|---|---|---|
| Untreated (5 mg/L) | 6 ± 5 | Untreated (1000 mg/L) | 1680 ± 75 |
| Treated for 7 min | 16 ± 5 | Treated for 40 min | 99 ± 5 |
| Treated for 15 min | 30 ± 5 | Treated for 120 min | 62 ± 5 |

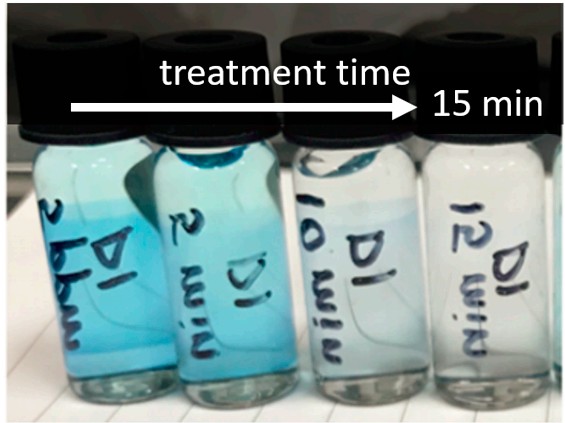 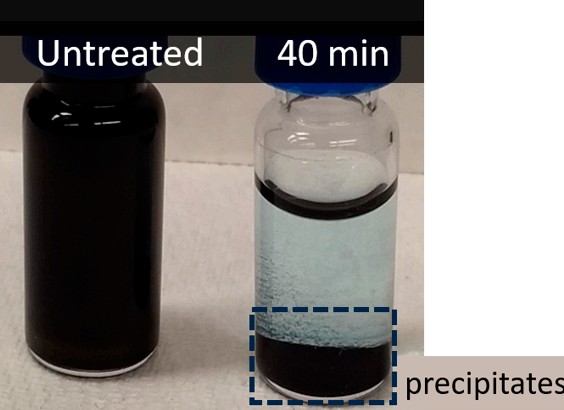

**Figure 3.** (**a**) Images of samples starting with 5 mg/L MB showing decrease in color indicating destruction of the MB dye. (**b**) Image of untreated sample with 1000 mg/L MB and sample after 40 min of plasma treatment showing MB precipitates and mostly clear supernatant.

The two primary long-lived plasma-derived species that could be present in the sample after plasma treatment are hydrogen peroxide and ozone, since nitrates and nitrites should not be generated under steam-mode operation. Of these, hydrogen peroxide has been found to interfere with the standard COD assessment by reducing the potassium dichromate oxidizing agent according to the overall oxidation reaction given in Equation (4) [31]. From this equation, the theoretical hydrogen peroxide interference is calculated to be 470.6 mg of COD as $O_2$ per 1000 mg $H_2O_2$.

$$K_2Cr_2O_7 + 3H_2O_2 + 4H_2SO_4 \rightarrow K_2SO_4 + Cr_2(SO_4)_3 + 7H_2O + 3O_2 \tag{4}$$

Researchers investigating plasma treatment of wastewaters must be particularly careful when interpreting COD results because the interference from residual hydrogen peroxide may not be evident. For example, two 100 mL samples of 1000 mg/L MB were prepared and treated with the underwater DBD plasma jet. After treatment, the COD decreased from the initial value of 1680 ± 75 mg/L, as shown in Table 1, and the supernatant becomes transparent as oxidized MB precipitates out of the solution, as shown in Figure 3b. This case of high initial COD relative to the residual peroxide concentration masks the peroxide interference effect. Still, any residual peroxide left in the sample after treatment will contribute to the COD. Hence, the COD contribution from organics should be less than the 99 mg/L after 40 min of treatment indicated in Table 1, since there is some contribution from residual peroxide.

In this case of high initial dye concentration (MB-B Sample, Table 1), the residual peroxide contribution to the COD may have been up 99 mg/L, but since the COD decreases from 1680 mg/L to 99 mg/L, it appears that the process is working as intended (COD decreases as MB is oxidized). In this case, the COD of the residual peroxide is at least 17 times less than the COD of the organics present in the sample before treatment, thus the interference effect is not immediately apparent. In reality, the treatment process is likely working better (faster organic removal rate) than would be indicated by the COD test because of the interference from residual peroxide. However, in the case of low initial dye concentration (MB-A Sample, Table 1), the measured COD increases with plasma treatment time

because the COD contribution from residual peroxide after treatment is greater than the COD of the organics present in the sample before treatment. By simply looking at the measured COD before and after treatment, it would appear as if the process did not work (i.e., organics were not removed), because the peroxide interference has not been corrected for.

To investigate the magnitude of hydrogen peroxide interference in COD assessments, de-ionized water was dosed with hydrogen peroxide both by solution and by DBD plasma jet according to the procedure described in Section 2.2. The hydrogen peroxide concentration is measured immediately before COD assessment. In both cases, the COD increases with increasing hydrogen peroxide concentration, as shown in Figure 4, even though there were no organics present in the samples. This proportional relationship was also found by Lee et al. for conventional advanced oxidation processes, although the interference magnitude varied depending on the water quality and treatment process [20].

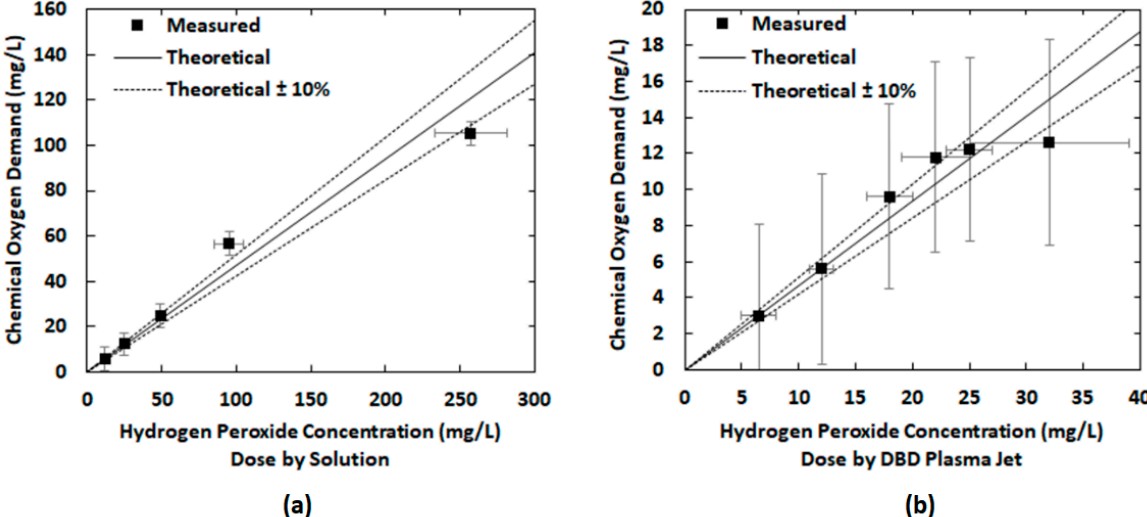

**Figure 4.** (**a**) COD vs. $H_2O_2$ concentration, where $H_2O_2$ is added to samples by a solution of 30% $H_2O_2$. (**b**) COD vs. $H_2O_2$ concentration, where $H_2O_2$ is added to samples by exposure to the underwater DBD plasma jet.

These results seem to confirm that the measured COD results from residual peroxide reacting with the potassium dichromate oxidizing agent during the COD assessment. The magnitude of the interference found in this experiment is approximately equal to the theoretical value from Equation (4), but this value can vary in practice by up to 16% percent depending on the specific organic compounds and oxidizing species that are present in the sample being assessed, particularly in real wastewaters [20]. When AOPs are used to remove trace levels of contaminants from relatively low COD waters, as can occur at final disinfection stages of drinking water treatment plants, the uncertainty in the actual interference magnitude can lead to misinterpretation of results. While reagents may be used to quench residual hydrogen peroxide in attempt to remove this interference, these reagents often introduce their own interferences with the COD assessment [31]. More research is needed to investigate how the residual oxidants and other long-lived species from plasma treatment affect the COD measurement in various synthetic and real wastewaters in order to develop correction methods for appropriate interpretation of results.

## 4. Conclusions

Wastewater treatment plants use COD to measure treatment process efficiency and quality of treated water. Plasma-water interaction produces residual peroxide which interferes with the COD measurement. As a result, COD assessments of plasma treated waters will show incomplete removal

of oxygen demand and may be viewed as a disadvantage of plasma-based advanced oxidation for municipal wastewater treatment plants and companies looking to treat their wastewater streams. Indeed, when researching new wastewater treatment technologies, LC-MS should be used to determine residual organic content before applying standard water quality diagnostics. This paper therefore describes precautions necessary for plasma practitioners to take into account so as to yield credible contaminant decomposition measurements. Plasma in contact with water produces a host of reactive oxygen species, and so caution is necessary when interpreting the results.

**Author Contributions:** Investigation, analysis, and writing, J.G.; conceptualization, resources, and supervision, J.F.

**Funding:** This research was funded by the National Science Foundation (NSF 1700848) and the U.S. Department of Energy (DOE DE-SC0001939).

**Conflicts of Interest:** The authors declare no conflicts of interest.

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
