# Peer review of "Hydrogen Peroxide Interference in Chemical Oxygen Demand Assessments of Plasma Treated Waters"

_plasma, doi:10.3390/plasma2030021_

Round 1

Reviewer 1 Report

-Various sections can benefit from some rewording. Example: Line 23:  demand is rising.. to meet needs?

- Line 40: The generation of these is dependent on the discharge, discharge temperature, etc. What type of plasma discharges are you referencing for each of these effects?

- Line 55 -57 are better, alluding to the dependency of the generation of effects on various plasma parameters. Still, line 61 again, makes a general statement about ozone that may not even be present dependent on the plasma parameters

- Line 66: As comments are made on residual hydrogen peroxide; authors should strongly consider review of references which clearly discuss the lifetime of H2O2 in plasma-treated water and water in general.

Line 134: Authors again mention operating in “steam mode.” Is there a different condition in which authors operated the set-up? Is the gas flow mode the opposite of what you have escribed as  steam mode?

-        Line 143: What is the order of magnitude of this “short delay?”

-        Line 162 Authors state that H2O2 is quantified prior to COD assessment

-        Line 193 – 194: Researchers should consider re-analyzing the samples at time scales aft the initial set of assessments: 1 min; 1 hour; 1 day. In such experiments, the presence or lack thereof H2O2 should be reported.

-        Overall, interesting the paper does a good job of explaining the phenomenon observed and making attempts to categorize and isolate some of the potential causes. Still, additional considerations should be made in this assessment, including observing changes in these values over time. Additionally, it is noted that these assessments are more susceptible to these error at lower initial concentrations; what is the cut-off point? Additional concentrations should be tested to properly categorize this phenomenon. Overall, the paper is a warning for those utilizing COD assessment after AOP treatment. Does plasma exacerbate this effect substantially? If so, why?

-        All the best with your research!

Author Response

Please see the attached Word document for a point-by-point response to the reviewer's comments.  

Reviewer 2 Report

Unfortunately, the scientific contributions from this manuscript are minimal/non-existent. From plasma perspective, nothing new is learned. From water treatment perspective, the same, in addition to text being written very simplistically. The main conclusions are that H2O2 interferes with the COD measurements with or without plasma as long as some H2O2 is present. That’s already known. Very limited data are presented and not well explained. Methodologies are not clear.

Minor comments:

The first word in the abstract is misspelled

The introduction section is written somewhat simplistically and repetitively. The paragraph 68-78 especially should be revised. TOC description is particularly unnecessary: lines 79-89. Also, the quality of writing should be improved.

The authors write: “The problem with COD assessments of plasma treated wastewaters is that residual peroxide interferes with the measurement. “ Can you please add references?

Experimental setup: am I understanding this correctly that the discharge takes place in air? Why not use helium or argon?

Do you measure nitrate and nitrite ions in the plasma-treated sample after 20 h?

Are you confident you are mineralizing the dye in 10 min? Did you measure the TOC? Yes, you are destroying the chromophore but not the dye organic structure itself.

If nitrogen-based compounds cannot be generated during the plasma process, how can ozone be generated?

Did you measure H2O2 generated with and without the dye? How much H2O2 can you actually generate in 10 mins?

How do you explain the results in Table 1 where COD decreases for high dye concentrations?

Line 203: what do you mean by dissolved MB fragments? Is this carbon? If so, what is the plasma temperature?

Line 207: you mention 40 minutes of treatment and earlier 10 minutes? Can the experimental conditions be clarified?

Your error bars on Figure 4 b are huge!

If others have already seen (Lee et al) what you are showing as the major result of this paper, how are you contributing to the knowledge?

Author Response

(The authors gave the same response as above.)

Round 2

Reviewer 1 Report

Again, overall good job at explaining your observations.

All the best with your research.

Reviewer 2 Report

Even though the authors provided explanations for my comments, the manuscript is essentially the same. I understand the message the authors are trying to convey, but the manuscript is written in a confusing way and the experiments executed somewhat poorly. There is an obvious lack of knowledge in the water chemistry which what this manuscript is about.